# Potential Anticancer Activity of Pomegranate (*Punica granatum* L.) Fruits of Different Color: In Vitro and In Silico Evidence

**DOI:** 10.3390/biom12111649

**Published:** 2022-11-07

**Authors:** Maria C. Cortez-Trejo, Francisco J. Olivas-Aguirre, Elisa Dufoo-Hurtado, Raquel Castañeda-Moreno, Hassan Villegas-Quintero, José L. Medina-Franco, Sandra Mendoza, Abraham Wall-Medrano

**Affiliations:** 1Programa de Posgrado en Alimentos del Centro de la República (PROPAC), Research and Graduate Studies in Food Science, School of Chemistry, Universidad Autónoma de Querétaro, Santiago de Querétaro 76010, Querétaro, Mexico; 2Department of Health Sciences, Campus Cajeme, University of Sonora, Ciudad Obregón 85199, Sonora, Mexico; 3DIFACQUIM Research Group, Department of Pharmacy, School of Chemistry, Universidad Nacional Autónoma de México, Mexico City 04510, Mexico; 4Instituto de Ciencias Biomédicas, Universidad Autónoma de Ciudad Juárez, Ciudad Juárez 32310, Chihuahua, Mexico

**Keywords:** cancer, foodinformatics, MTT, nutraceuticals, phenogenotype, phytochemicals, pomegranate, *Punica granatum*

## Abstract

Pomegranate (PMG; *Punica granatum* L.) fruits possess a well-balanced nutrient/phytochemical composition, with proven adjuvant benefits in experimental cancer chemotherapy; however, such bioactivity could be affected by PMG’s phenogenotype (varietal). Here, the chemical and phytochemical (UPLC-DAD-MS^2^) composition, antioxidant capacity and anticancer potential [in vitro (MTT assay) and in silico (foodinformatics)] of three PMG fruits of different aryl color [red (cv. Wonderful), pink (cv. Molar de Elche), and white (cv. Indian)] were evaluated. The macro/micronutrient (ascorbic acid, tocols, carotenoids), organic acid (citric/malic), and polyphenol content were changed by PMG’s varietal and total antioxidant activity (ABTS, alcoholic > hexane extract) in the order of red > pink > white. However, their in vitro cytotoxicity was the same (IC_50_ > 200 μg.mL^−1^) against normal (retinal) and cancer (breast, lung, colorectal) cell lines. Sixteen major phytochemicals were tentatively identified, four of them with a high GI absorption/bioavailability score [Ellagic (pink), vanillic (red), gallic (white) acids, D-(+)-catechin (white)] and three of them with multiple molecular targets [Ellagic (52) > vanillic (32) > gallic (23)] associated with anticancer (at initiation and promotion stages) activity. The anticancer potential of the PMG fruit is phenogenotype-specific, although it could be more effective in nutraceutical formulations (concentrates).

## 1. Introduction

Cancer is one of the five leading causes of adult death worldwide, and its burden is rising in a disproportionate manner. Data from the global cancer observatory (GLOBOCAN-GCO) indicate that ~10 million deaths and ~19.3 million new cases occurred in 2020, with breast (11.7%) > lung > colorectal > prostate > stomach (5.6%) cancers the most diagnosed and lung (18%) > colorectal > liver > stomach (6.9%) cancers the most lethal [1,2]. Cancer imposes the highest clinical, social, and economic burden in terms of cause-specific disability-adjusted life years (DALYs) among all human diseases [3]. While acknowledging that the most effective way of reducing cancer risk is targeting preventable risk factors (e.g., unhealthy nutrition, smoking) and correct diagnosis/control of non-preventable ones (e.g., genetic factors and primordial inflammatory response), the repositioning of OTC drugs not commonly used as chemotherapeutic aids (e.g., metformin) and the use of natural bioactive compounds (e.g., antioxidants) are effective primary/secondary aids [4,5]. It is noteworthy that several cross-sectional and prospective epidemiological studies have documented the value of fruit-rich diets in reducing the odds of certain types of cancer [6,7].

Pomegranate [PMG; *Punica granatum* L., Latin: *pomum* (apple) and *granatus* (grainy)] is considered a “Super Fruit” due to its excellent nutritional (macro/micronutrients) and functional (bioactive xenobiotics) health-promoting profile [8]. The list of PMG phytochemicals is quite vast, but polyphenols have been the most studied and can be found in PMG´s fruit (aryl; mesocarp), peel (exocarp), seed, flower, bark, flowers, and leaves [8,9]. The chemical structure, molecular properties, and physiological mechanisms associated with the oncosuppressive activity (e.g., cell cycle arrest at S-G_2_-M and, pro-apoptosis, free-radical inhibition/trafficking, direct damage of DNA, antiangiogenesis, antimigration/invasion) of PMG´s polyphenols, have been studied in detail for breast, lung, thyroid, colon, and prostate cancer [7,8,9,10]. However, PMG fruits are genetically diverse. This characteristic impacts upon many phenotypical traits (e.g., fruit color size, flavor), the phytochemical composition, and the physicochemical characteristics, thereby affecting their bioactive capacity. For instance, despite the limited (disease-specific) cheminformatic (*in silico*), pre-clinical (in vitro/in vivo) and clinical (human trials) evidence, strong-to-light colored PMG varietals seem to follow a high-to-low bioactivity trend [8], although such a trend may vary among human diseases including cancer types [5,10,11].

The aim of this study was to associate the chemical/phytochemical profile of three PMG fruits of different flesh (aryl) colors [red (cv. Wonderful), pink (cv. Molar de Elche), white (cv. Indian)] with their in vitro/in silico anticancer potential. To our knowledge, the comprehensive [in vitro (several cell lines, high-throughput chemical characterization) plus in silico (cheminformatics)] screening for the genotype-specific anticancer potential of PMG fruits, is here reported for the first time; from the computational point of view, the application to the informatics methods involved in food chemical research (foodinformatics) is innovative.

## 2. Materials and Methods

### 2.1. Reagents

Pure (≥93%) standards [(+)-carotenoids, tocols (tocopherols + tocotrienols), phenolic compounds, and Trolox], radicals (DPPH, ABTS▪^−^, AAPH), Folin–Ciocalteau phenol reagents, fluorescein, and ACS grade salts and acids were obtained from Sigma-Aldrich-Fluka (St. Louis, MO, USA) or Cayman Chemical (Ann Arbor, MI, USA). All HPLC-MS or analytical-grade solvents were obtained from JT-Baker (Avantar Performance Materials S.A. de C.V., Mexico, Mexico). Cell culture materials [Dulbecco’s modified Eagle’s medium (DMEM), 10% fetal bovine serum (FBS), 1% glutamine, 1% penicillin, dimethyl sulfoxide (DMSO), 3- (4,5-dimethylthiazol-2-yl) 2,5 diphenyltetrazolium bromide (MTT), Trypsin EDTA] were purchased from Sigma-Aldrich-Fluka (St. Louis, MO, USA). Conversely, human cell lines [Normal retinal (ARPE-19; CRL-2302 TM), breast ductal carcinoma (T-47D; HTB-133 TM), breast adenocarcinoma (MDA-MB-231; HTB-26 TM), alveolar adenocarcinoma (A549; CCL-185 TM) and colorectal Duke’s type II adenocarcinoma (LS180; CL-187 TM)] were purchased from ATCC (ATCC^®^, Rockville, MD, USA).

### 2.2. Pomegranates (PMG)

Three varieties of ripped (14–18 ^o^Brix) PMG fruits, differing in their flesh (aryl) color [“Wonderful” (red), “Molar de Elche” (pink), and “Indian” (white)], were evaluated. PMGs were either purchased in the local market or harvested from a location nearby Hermosillo, Sonora, Mexico (29°10′56′′ N–110°52′54′′ W, 250 m ASL) and transported under refrigeration conditions (4 °C) immediately to the laboratory. Arils (mesocarp) were manually separated from PMG peel (pericarp + calyx), cleaned under aseptic conditions, air-dried in an adsorbent cloth (in the dark), freeze-dried (48 h, 1.2 L light-protected flasks; LABCONCO Freezone 6 freeze-dryer, Labconco, Kansas City, MO, USA), ground to a fine powder (≤40 μm), and stored under vacuum conditions at −80 °C in vacuum-sealed bags until use. Freeze-dried organic extracts from these were also stored at −80 °C for chemical, HUPLC-MS/MS, and bioactivity analyses.

### 2.3. Chemical Analysis

Triplicates analyses (CV ≤ 10%) were performed in all three dried PMG samples, using official AOAC methods for ash, protein, fat, moisture, and carbohydrate (by difference) contents.

### 2.4. Non-Phenolic Antioxidants

Carotenoids [α-carotene, all-trans-β-carotene, cryptoxanthin, lutein, and zeaxanthin] were extracted (methanol + tetrahydrofuran) from all dried PMG samples and quantified (μg. 100 g^−1^) by HPLC-DAD (450 nm) as recently described [12]; ascorbic acid (mg/100 g DM) was extracted (HPO3: glacial acetic acid: H_2_O) and quantified (μg. 100 g^−1^) by normal phase-HPLC as recommended by Doner and Hicks [13]. Tocols [Tocopherols (T) and tocotrienols (T3), α, β, γ, δ] were stepwise-extracted (hexane–metanol) and quantified (μg. 100 g^−1^) by normal-phase HPLC as recently described [14]. Pure standards were used to build calibration curves, and inter-run variation was ≤15%.

### 2.5. Phenolic Compounds (PC)

PC were extracted from 1 g of sample with either H2O–methanol (20:80 *v*/*v*; 20 mL; total phenols, total flavonoids) or acidified H2O–methanol [20:80 *v/v* with 2% *v/v* formic acid; 20 mL; monomeric anthocyanins (ANTO)], centrifuged (3000 rpm, 15 min, 2–5 °C), filtered (0.22–0.45 µM) and quantified [Total phenols (675 nm), mg of gallic acid equivalents (GAE). 100 g^−1^ dwb; total flavonoids (510); mg of quercetin equivalents (QAE). 100 g^−1^ dwb; ANTO (520, 700 nm); mg of Cyanindin-3-O-glucoside (Cy3GE). 100 g^−1^ dwb] in a FLUOstar Omega multifunctional microplate reader (BMG LABTECH Inc., Durham, NC, USA), according to standard colorimetric methods [15,16,17].

### 2.6. UPLC-DAD-MS^2^

Samples (1 g) were extracted [10 mL, H_2_O (1.8)/formic acid (0.2)/methanol (8.0)], sonicated (30 min), centrifuged (10,000 rpm, 4 °C, 25 min) twice under dark conditions, and combined supernatants (*n* = 2) were filtered (0.22–0.45 µm). Individual PCs were identified by HUPLC-DAD-MS^2^ using a Vion IMS QT of the mass spectrometer (HUPLC Acquity Class I, 2.1 × 100 mm, 1.7 µm BEH C_18_ column; Waters Corp., Milford, MA, USA) according to Amaya-Cruz et al. [18]. Running conditions were as follows: column temperature (35 °C), mobile phase = solvent A (H_2_O (99.9)-formic acid (0.1) and solvent B (acetonitrile), flow rate (400 μL. min^−1^), gradient = 5% B (2 min), stepwise increment ran up to 95% B (2–22 min) and then was held for 3 min, acquisition [positive (ANTO) and negative (all other PC) mode; of 50–1800 Da at 260, 280, 320, 360, and 520 nm], capillary energy (3.5 kV), low collision energy (5 eV), and high energy ramp (15 to 45 eV), mass calibration standard (leucine-enkephalin; infusion of 10 µL.min^−1^, 556.677 *m*/*z*), source temperature (120 °C), gas (Ar) flow (800 L.h^−1^). Tentative chemical identity was assigned by comparing their retention times (*rt*) with those reported in the literature, and by the mass fragmentation pattern ([M]*_m_*_/*z*_, fragments), using the UNIFI scientific information system and the Progenesis QI software (Waters Corps., Milford, MA, USA). Results were expressed as a relative abundance (%) to the total chromatogram peak area per PMG sample.

### 2.7. Antioxidant Activity

The radical scavenging (RSC) of DPPH (at 518 nm) and ABTS (at 734 nm) radicals were performed according to Brand-Williams et al. [19] and Re et al. [20], respectively. The ORAC assay was performed according to Ou et al. [21] using fluorescein [10 nM, excitation (485 nm)/emission (520 nm)], AAPH (240 mM) and Trolox (0.006–0.2 μmol/mL, R2 ≥ 0.95) as standard. All values were expressed in terms of micromole of Trolox equivalents (TE) per gram of dry weight (µmol TE/g dw). For lipophilic fractions, the oils were diluted in 2-propanol (5% *v*/*v*) and 1-butanol (1:25 *v*/*v*) for DPPH and ABTS assays, as suggested by Christodouleas et al. [22]. Results were expressed as micromoles of Trolox equivalents per gram of oil (µmol TE.g^−1^ oil).

### 2.8. Foodinformatics

Major PMG phytochemicals [*n* = 16; ellagic acid (1), ellagic acid glucoside (2), ellagic acid-4-O-xylopiranoside (3), gallic acid (4), galloyl-6-O-glucoside (5), 1,3,6-Tri-O-Galloyl-D-Glucose (6), vanillic acid (7), D-(+)-catechin (8), phlorozin (9), phellatin (10), quercitrin (11), citric acid (12), cyanidin-3,5-O-diglucoside (13), Del-phnidin-3-O-glucoside (14), Delphinidin-3,5-O- diglucoside (15), and pelargonidin 3,5-O-diglucoside (16)] with the highest relative abundance (UPLC-DAD-MS^2^) were evaluated. Cheminformatics and bioactivity prediction data (including ADMET properties) were calculated using canonical SMILES (Simplified Molecular Input Line Entry Specification) sequences retrieved from the PubChem server (https://pubchem.ncbi.nlm.nih.gov/; accessed date: 6 October 2022) and using well-established, validated, and public servers: admetSAR 2.0 (http://lmmd.ecust.edu.cn/admetsar2; accessed date: 15 August 2022), SwissADME (http://www.swissadme.ch/index.php; accessed date: 15 August 2022) [23], and Molinspiration (https://www.molinspiration.com/cgi-bin/properties; accessed date: 15 August 2022).

Bioavailability radars and boiled-egg diagram [WLOGP (atomistic interpretation of the fragmental system of Wildman and Crippen: lipophilicity) vs. TPSA (topological polar surface area: apparent polarity) plot] were used to infer the gastrointestinal (GI) absorption and blood–brain barrier (BBB) permeation of all sixteen compounds. Two-dimensional chemical features were used to characterize the chemical space. The chemical space analysis focused on four physicochemical properties (PCP) of pharmaceutical relevance [octanol/H_2_O partition coefficient (cLog P), molecular weight (MW), TPSA, and solubility (Ali/ESOL)]. PCP-based clustering depicted as 3D-dynamic plots (PC1-PC3) was generated with DataWarrior version 5.2.1 [24]. Potential protein targets (cancer-related, other diseases) for each compound were predicted with SwissTargetPrediction 2019 (http://www.swisstargetprediction.ch/; accessed date: 15 August 2022), and their linkage with cancer and other diseases was confirmed with UniProt [25] and Pharos [26] databases.

### 2.9. In Vitro Cytotoxicity

H_2_O-MeOH (20:80 *v*/*v*) extracts were obtained from each PMG sample (1:5 *w*/*v*), supernatants were rotoevaporated (40 °C) and then freeze-dried (as described above) before being further dispersed in DMSO (stock 80 mg.mL^−1^) and with DMEM to desired concentrations (12.5–200 μg.mL^−1^). The viability of human normal retinal (ARPE-19; CRL-2302 TM), breast ductal carcinoma (T-47D; HTB-133 TM) and breast (MDA-MB-231; HTB-26 TM), alveolar (A549; CCL-185 TM), and colorectal Duke’s type II (LS180; CL-187 TM) adenocarcinoma cell lines was assayed by the MTT assay as previously described [27] and tracked for 24–48 h. Only cells with >90% viability, passage number <10 and in the log growth phase were used. The absorbance [570 (test)/650 (reference) nm] of formazan crystals was measured with an ELISA plate reader (Multiskan EX, ThermoLabSystem, Cergy Pontoise, France), and results were recorded as the percentage of cell viability. Additionally, the half-maximal inhibitory concentrations (IC_50_) were further calculated. IC_50_ values < 10, 11–99 and 100–500 μg.mL^−1^ are customarily considered as displaying very strong, strong, and moderate cytotoxicity [28].

### 2.10. Statistical Analysis

All continuous variables were tested for normality (Shapiro–Wilk test) and homogeneity of variance (Levene´s test) and expressed as mean ± standard deviation (SD). One-way ANOVA followed by Tukey’s post hoc test were carried out to evaluate any significant differences (*p* < 0.05) between PMG samples. Pearson product–moment correlation coefficients (r) were calculated (and depicted as heatmap and PCA plots) to detect linear relationships between antioxidant phytochemicals (phenolic and non-phenolic) and overall antioxidant capacity (hydroalcoholic and hexane extracts). Statistical differences were considered as being *p* < 0.05. Data analysis was performed using NCSS 2000 (NCSS Statistical Software, Kaysville, UT, USA).

## 3. Results and Discussion

### 3.1. Chemical Profile of PMG Samples

The macronutrient composition of freeze-dried red (cv. ´Wonderful´), pink (cv. ´Molar de Elche´), and white (cv. ´Indian´) PMGs is reported in Table 1. All three samples showed slight differences [ red (↑ carbohydrates), pink/white (↑ all other macronutrients)] in their mean composition [g.100 g^−^^1^: Carbohydrate (70–79) > water > protein > ash > fats (0.4–0.9] and energy content (~314–342 kcal. 100 g^−^^1^).

This composition is quite like that of USDA Food data central (FDC)-registered raw PMG (FDC ID: 169134, NDB number: 9286; https://fdc.nal.usda.gov/index.html; accessed date: 15 August 2022), and to other PMGs from India [29], and South Africa [30]. Even though the chemical composition of PMGs can be affected by several pre/post-harvest factors, even for the same phenogenotype (varietal). Its nutrient density is higher than its energy density in such a way that the fruit can be recommended for primary (healthy eating) and secondary (functional macro/micronutrient profile) prevention of many diseases, including cancer [8,9,10].

### 3.2. Non-Phenolic Antioxidants

One of the many mechanisms by which PMG fruit´s phytochemicals exert their anticancer effects is through antioxidant activity. The total antioxidant capacity of PMG is mainly driven by phenolic species (to be discussed in the Section 3.3), ascorbic acid, carotenoids and tocols (T + T3), and other free radical scavengers with anticancer potential [5,14,23]. According to Table 1, white (´Indian´) PMG is richer in ascorbic acid, α, β-carotenes, γ-T, and β-T3 than red (´Wonderful´) and pink (´Mollar de Elche´) PMGs, but the later are better sources of lutein and total carotenoids; interesting, total tocols were found in white > pink > red PMG. Similar results in ascorbic acid content for white vs. red PMG were reported by Opara et al. [31], also agreeing with Costa et al. [32] who reported γ-tocopherol as the main vitamin E (tocol) isoform in PMG seed oils from Turkey and Israel. It is noteworthy that PMG aryls contain both seeds and juice (here combined in PMG samples) whose non-phenolic antioxidant phytochemicals contribute to the stronger hydroxide RSC and better DNA damage-preventing capacity of white PMG as compared to red PMG [33], which usually exhibits much more antioxidant capacity and phenolic content/diversity. It should be noted that, although it was not evaluated in our study, the contribution of vitamin C, carotenoids, and tocols to the total RSC should be expected to be lower than that of phenolic compounds, which are in higher concentration and diversity in the PMG samples [27].

Several epidemiological (cross-sectional/longitudinal) studies, systematic reviews and meta-analyses have supported the benefits of vitamin A (+ carotenoids), C (ascorbic acid), and E (T + T3 isoforms) in the primary prevention of certain human cancers [5]; however, evidence on the benefits of these vitamins as adjuvant chemotherapy aids remains questionable. For example, ascorbic acid improves the immunological status of patients receiving intensive chemotherapy and/or stem cell transplantations, particularly when administered parenterally rather than orally, although other effects (e.g., overall survival, clinical status, quality of life, and performance status) are not evident [34]. Carotenoid pigments are also considered potential candidates for chemoprevention and chemotherapeutics of breast, colorectal, lung, and prostate cancers and their cytotoxic/antiproliferative effect is related to a plethora of molecular mechanisms in a cancer-type manner [35]; however, such bioactivities are firstly conditioned by their first/second-pass metabolism when orally administered [5]. As for vitamin E [tocols (T + T3)], certain isoforms (γ-T, δ-T, γ-T3, δ-T3) have antiproliferative/pro-apoptotic effects on many types of cancer cells in vitro and suppress tumor progression in vivo (preclinical cancer models) by modulating key signaling pathways/cancer (e.g., eicosanoids, NF-κB, STAT3, PI3K, and sphingolipids, although such relationship is conditioned to their bioavailability) [36]. Nevertheless, although PMG aryls are not particularly rich in carotenoids and tocols, and although they are in ascorbic acid, particularly sour PMGs, their differential/synergistic contribution to the oncosuppressive activity of PMGs of different phenogenotypes remains to be evaluated [7,8,9,10].

### 3.3. Phenolic Compounds (PC) and Organic Acids

PC are a group of simple-to-complex molecules, characterized by having at least one phenolic ring in their structure. The bioactivity of PC in cancer is not only limited to their role in cell defense (e.g., antioxidant and anti-inflammatory activity) but also by exerting specific mechanisms at early cancer stages (initiation–promotion–transformation–progression), examples of which include cell signaling, epigenetic action, hormonal/enzyme control, immunoenhancing activity, etc. PC bioactivity depends on the chemical structure, the dose/route at which they are administered, and their synergistic/antagonistic effects with other biomolecules [5,14].

According to Table 1, total (1820–2450 mgGAE.100 g^−1^ dwb) and subclass [flavonoids (170–320 mgQE.100 g^−1^ dwb), ANTO (0–8070 mgCy3GE.100 g^−1^ dwb)] PC content in PMG samples were ordered in red > pink > white, also coinciding with the relative abundance of phenolic acids and derivates (*n* = 7; Major: galloyl 6-O-glucoside), flavonoids and derivates (*n* = 10; Major: phellatin, quercitrin), organic acids (*n* = 2: citric > malic) and ANTO (*n* = 4; Major: Cyanidin 3,5-O-diglucoside), as determined by UPLC-DAD-MS^2^ (Appendix A). Most identified phytochemical species have been previously reported for PMG phenogenotypes from Italy (37, 38), Croatia (39), Spain (40), and the USA (41). It is noteworthy that the PC profile (molecular diversity and content) of PMGs depends on the fruit´s ripening stage, phenogenotype, pre/post-harvest conditions, and extractive solvent, among other factors [37]. Nevertheless, the flavonoid: total PC and ANTO: flavonoid ratios and ANTO fingerprint (cyanidin > delphinidin > pelargonidin, di > mono glucosides) are quite conserved among PMGs of different phenogenotypes and regions [8,37,38,39,40,41]. Also, while red PMG was a richer source of ANTO and organic acids, white and pink were richer in phenolic acids (Appendix A).

Ellagitannins and ANTO have been the most investigated PC in PMGs because of their well-known antioxidant, anti-inflammatory, cardioprotective and anticancer properties [42,43,44]. Particularly, ellagitannins [43] and ANTO [44] have potent antiproliferative and pro-apoptotic activity against several cancer cells including those of the colon, breast, lung, cervix, prostate, liver, and oral/esophageal, to name a few. Although the extractive methodology and chromatographic conditions used in this study did not allow the identification of complex ellagitannins (e.g., punicalagin), as it has been reported for “Wonderful” and “Mollar de Elche” varieties by others [37,38,39,40,41] that the presence of ellagic acid and two of its glycosides in all three varieties of PMG (pink > red, white; Appendix A) accounts for this. PMG´s ellagitannin-derived compounds (e.g., ellagic and gallagic acids, and urolithins A/B and B) not only have a high RSC but also antimutagenic activity [42,45]. Gallic acid and its glucosides (present in pink/white > red PMG; Appendix A) and gallotannins are also bioactive against many cancer cells, including abnormal colonocytes [46]. Lastly, even though monomeric ANTO in white PMG fall below detection/quantification limits with the pH-differential spectrophotometric method [17], it was possible to identify them by UPLC-DAD-MS^2^, evidencing the lack of sensitivity/accuracy of the spectrophotometric method.

According to Nicolson et al. [47], PMG small flavonoids (e.g., catechin, epicatechin, quercetin, naringenin) have strong electron-acceptor properties, making them potent inhibitors of the NF-κB pathway involved in several carcinogenic processes. In this study, the five most abundant flavonoids were phellatin, quercitrin, phlorizin, D-(+)-catechin, and corilagin (Appendix A), although each PMG sample had a differential (relative abundance) flavonoid fingerprint: red [quercitrin> (-)-epicatechin], pink (phellatin, > corilagin, > astragalin), and white [D-(+)-catechin, phlorizin]. Considering their differential flavonoid fingerprint and the intrinsic physicochemical features of each major molecule (Appendix A), different anticancer mechanisms (e.g., anti-inflammatory, apoptosis/autophagy, chemical affinity to cancer stem cells, antiangiogenic/antimetastasis, etc.) and chemotherapeutic action [48] could be expected for PMG samples. Lastly, citric and malic acids, the major organic acids in PMG juice [40,42] and in the assayed PMG samples, have been reported as suppressing A549 lung cells by a feedback control over cell glycolysis and TCA cycle [49].

### 3.4. Antioxidant Capacity

As previously stated, antioxidant phytochemicals play an outstanding role in chemoprevention [5], including DNA-protective capacity [33], as well as in enhancing anti-inflammatory and other cytoprotective effects [8,11,42,44]. According to Figure 1, the radical assay sensitivity to detect differences between samples was ABTS > DPPH > ORAC in methanolic (hydrophilic antioxidants) hexane (hydrophobic antioxidants) extracts; the ABTS▪− radical was efficiently scavenged (µmol TE/g dw) by red (44.5) > pink (36.9) > white (19.6) PMG in methanolic extracts (ABTS-M) and white (10.0) > red, pink (3.8) PMG in hexane (ABTS-H) extracts.

Pearson’s product–moment correlation analysis (Appendix A), evidenced a positive strong-to-moderate trend (r ≥ 0.74) between ABTS-M and total PC, flavonoids, carotenoids contents, and DPPH-M. However, ABTS-M demonstrated an inverse relationship (r ≥ −0.73) with ascorbic acid and tocol content, as well as the scavenging ability of DPPH and ABTS▪− radicals for hexanoic extract (DPPH-H and ABTS-H). Conversely, ABTS-H a showed strong positive (r ≥ −0.88; ascorbic acid, tocol and DPPH-H) and moderate (r ≥ −0.61) relationship with the rest of the assayed parameters. Yang et al. [50] studied thirteen PMG varieties cultivated in China (Zaozhuang), reporting a positive correlation between antioxidant capacity (DPPH radical) and total contents of PC, flavonoids, and hydrolyzable tannins (r ≥ 0.89) in both the pericarp (peel) and seeds (aryls with no juice), but not in juice. Similar results were also reported for Spanish cultivars, which also include “Wonderful” and “Mollar de Elch”’ [40].

The aforementioned apparent causal relationships (↑ [antioxidant species]: ↑ RSC; ↑ [hydrophilic antioxidants]: ↓ [lipophilic antioxidants]) are partially explained by the individual antioxidant capacity (radical-specific quenching) and the relative abundance of each antioxidant phytochemical extracted under hydrophilic (methanol—water; LogPow ~1.0) and hydrophobic (hexane; LogPow = 3.9) conditions [27]. Unlike DPPH and ABTS▪− radicals-based assays, the ORAC assay is considered more convenient to evaluate chain-breaking antioxidant activity within biological systems where reactive oxygen (ROS) and nitrogen species (NOS) are more relevant [20]. ROS plays a dual role in cancer by promoting cell proliferation–survival–adaptation to hypoxia and by triggering oxidative-stress cancer cell death, both cases handled with ROS-manipulation therapies [51]. However, this assay was less sensitive to discriminate PMG phenogenotypes, a fact that deserves a more detailed study aimed to uncover the fine-tuning mechanism exerted by each antioxidant specie.

### 3.5. In Silico Gastrointestinal Fate Prediction

Machine learning/artificial intelligence (ML/AI) algorithms have recently gained attention within the pharma/nutraceutical industry and are being applied at different stages of the natural product drug research pipeline [52], including those derived from foods. This staging research process is initially fed with either chemical structure data coming from high-through output chemical characterization methods (e.g., GC-MS, HPLC-MS) or by computer-assisted data mining of natural products [53]. The next step is to process and filter large chemical databases by using cheminformatic platforms, aiming to explore the bioactive potential of each molecule (targeted/untargeted approaches) and artificially ranking its target bioactivity and applications. Particularly, ADME/Tox (physicochemical properties) screening + untargeted bioactivity prediction (pharmaceutical interest) is probably the most useful cheminformatic approach in the nutraceutical field [23,24].

Here, sixteen (15 PC + citric acid) compounds were considered major PMG phytochemicals (Figure 2) from twenty-three bioactive compounds identified by UPLC-DAD-MS2 (Appendix A). The ADMTE/Tox profile was initially estimated to predict their GI-to-BBB fate (Appendix A) and their potential cancer-targeting effect. As previously described (Section 3.3), PMG samples ranked in the order of red (7), pink (6) > white (3) as major sources of phytochemicals (Figure 2). Those considered as GI-permeants (*n* = 4; bioavailability score = 0.55–0.85, synthetic accessibility = 1.22–3.50) do not violate the Lipinski’s rule of five (LR5V) as related to six core physicochemical features (Low MW/TPSA, smaller number of rotable bonds/H-bond/acceptor) when compared to GI non-permeants (*n* = 12; bioavailability score = 0.17–0.56; synthetic accessibility = 2.18–6.59; Appendix A). GI-permeants [Name (compound ID/PMG): Ellagic (1/pink), gallic (4/white), vanillic (7/red) acids, and D-(+)-catechin (8/white); Figure 3] also exhibited structure-specific reactivities toward the relevant biomolecules (Appendix A): P-glycoprotein (P-gp) substrate and G protein-coupled (GPCR)/nuclear receptor ligand, protease/enzyme inhibitor (8), cytochrome P450 inhibitor [(1) 1A2, (4) 3A4], pan assay interference (PAINS) compounds [Catechol-A (1, 4)] and molecules with leadlikeness (starting molecule for chemical optimization: 1, 8).

Bioavailability radars offer preliminary insights into the drug-likeness of natural bioactive compounds [23], and the structural feature distinguishing GI-permeants from non-permeants is the low fraction of carbons in the sp^3^ hybridization state. However, none of these sixteen molecules can cross the BBB (Figure 3). According to Daina et al. [23] small molecules (MW, 150–500 g/mol) with low TPSA (20–130 Å^2^), solubility (LogS ≤ 6.0), and flexibility (≤9 rotatable bonds), yet a high fraction (≥0.25) of sp^3^ carbons, are more likely to permeate GI/BBB. In this study, phenolic acids (aglycones) but not their glycosylated derivates (bioavailability score = 0.17–0.55, synthetic accessibility 3.79–5.32) complied with these chemical features. Phenolic acids (PC bearing at least one phenol moiety within a resonance stabilized structure) are efficiently extracted from food matrices within the GI tract (bioaccessibility) and effortlessly absorbed into the intestine, reaching systemic circulation quite rapidly [53]. Under simulated (in vitro) gastrointestinal conditions (static in vitro digestion), PMG phenolic acids are more stable in gastric and intestinal conditions than ANTO and Punicalagin A (ellagitannin) [54]. Moreover, glycosylated derivates and complex forms of phenolic acids (ellagitannins and gallotannins) can be efficiently hydrolyzed by pancreatic (hydrolases) or microbial (esterases) enzymes into their monomeric forms, which are further biotransformed into cinnamic (ellagic acids)/benzoic (phenolic acids) metabolites of much lower MW yet higher bioactivity, increasing their odds of BBB permeation [55]. The bioaccessibility and antioxidant activity of catechins (including D-(+)-catechin) positively correlates with their binding affinity to dietary (e.g., β-lactoglobulin, and β-casein) and GI proteins (e.g., P-gp) under simulated GI conditions, a fact particularly important in colon cancer chemotherapy [56,57].

### 3.6. Protein-Targetting Prediction

Almost every natural product has special chemical properties which endow it with the capacity to ameliorate various diseases. Particularly, PMG phytochemicals exert antioxidant, anti-inflammatory, and hypoglycemic effects associated with a plethora of non-communicable diseases, although their oncosuppressive action seems to be cancer-specific [58]. On the latter poiny, the antiproliferative (cell cycle arrest at S-G2-M), pro-apoptotic, free-radical inhibition/trafficking, antiangiogenic, and antimetastatic (antimigration/invasion) qualities of PMG’s PC have been studied in detail for breast, lung, thyroid, colon, and prostate cancer [7,8,9,10]. In this study, potential protein targets in cancer (initiation-promotion-transformation-progression) and other metabolic intermediaries were predicted with SwissTargetPrediction 2019. This is an open web tool that calculates the similarity threshold of a given small molecule with other molecules compiled in curated and cleansed collections of known bioactive compounds (reverse screening) [59], and those with presumably anticancer potential were confirmed with UniProt [25] and Pharos [26] databases. The number of cancer-related protein targets was PMG-specific [red (129) > pink (109) > white (55)], as related to their phytochemical fingerprints (Figure 4) but also bioavailable PC [Ellagic (52) > vanillic (32) > gallic (23) acids; D-(+)-catechin: none].

It is noteworthy that the number of protein targets against cancer followed the same trend as for other targets involved in other cellular processes, both in bioavailable (all but D-(+)-catechin) and non-bioavailable phytochemicals (Figure 4). Bioavailable phenolic acids [Figure 3; ellagic (1), gallic (4), and vanillic (7) acids)] have demonstrated several anticancer mechanisms in vitro, ex vivo, and in vivo [53,60,61].

Lastly, among other protein targets relevant in the clinical course of cancer, ellagic acid, gallic acid, and vanillic acid to a lesser extent, have a strong inhibitory activity on several human carbonic anhydrase (hCA, metalloenzyme; E.C. EC 4.2.1.1) isoforms (I, II, II, IV, VB, VI, VII, IX, XII, XIV). hCAs are strategically distributed in human organs and cells (cytosolic, membrane-associated, fluids) and are associated with key physiological activities; particularly, within the microenvironment of tumor cells, membrane-bound hCAs IX and XII activate the hypoxia-inducible factor 1 and 2 (HIF-1/2) consequently activating the expression of several genes involved in glucose metabolism, angiogenesis, and pH regulation [62,63]. Since natural products bearing phenolic rings are potent inhibitors of hCAs (in an isoform-specific manner) [64] and both ellagic and gallic acids have a catechol A-PAINs alert (Appendix A), this explains their promising potential as hCA inhibitors.

### 3.7. In Vitro Citototoxicity Assays

The cytotoxic activity (MTT assay) of PMG samples against normal (retinal) and cancer (breast, lung, colorectal) cell lines is depicted in Figure 5.

Despite an apparent red > pink > white trend (all but A549 cells; Figure 5), all PMG hydroalcoholic extracts are moderately active (IC_50_> 200 μg.mL^−^^1^) on the assayed cell lines since IC_50_ values < 10, 11–99 and 100–500 μg.mL^−^^1^ are customarily considered as very strong, strong, and moderate cytotoxic [28]. Moreover, at given concentrations, they exhibited growth-promoting effects in both normal ARPE [pink (12.5–50.0, ~11%), white (12.5, ~6%)] and cancer cell lines (LS180, 7–47D > A-549, MDA-MB-231). It has been reported that raw food-derived extracts sometimes induce proliferation in both normal and cancer cell lines, particularly at low concentrations [65,66,67]. The lack of bioactivity (IC_50_ > 200 μg.mL^−^^1^) of raw fruit extracts has been also reported for pecan nut in A549, LS 180 and ARPE-19 cells [65] and PMG juice in MDA-MB-231 [66]. Conversely, pure gallic acid from PMG (cv. Kabul; IC_50_ ~ 15 μg.mL^−^^1^) and mango (cv. Ataulfo; IC_50_ ~ 50 μg.mL^−^^1^) peels inhibit A549 and LS180 cell growth [46,67]. The observed low cytotoxic effect (IC_50_ > 200 μg.mL^−^^1^) can be ascribed to low extract concentration, short exposure time, and the co-existence of cell growth-promoting substrates (e.g., free sugars). Lastly, certain cellular events suggestive of cytotoxicity (e.g., pro-apoptotic events such as cell shrinkage/rounding, nuclear fragmentation/chromatin condensation, membrane ‘blebs’) could have occurred at the upper assayed concentration (200 mg.dl^−1^); these were not documented in this study (e.g., by optical microscopy), a fact that deserves to be explored soon.

## 4. Conclusions

In this study, the in vitro assays indicated that the antioxidant capacity (ABTS, alcoholic > hexane extract) and phytochemical fingerprint [ascorbic acid, tocols, carotenoids, organic acids, and PC (relative abundance/chemical nature)] of PMGs is phenogenotype-specific [red > pink > white], although their in vitro cytotoxicity was the same (IC_50_ > 200 μg.mL^−1^) against normal (retinal) and cancer (breast, lung, colorectal) cell lines. On the other hand, chemometric analyses showed that ellagic (pink) > gallic (white) > vanillic (red) acids may interact with multiple molecular targets in the cancer continuum (initiation–progression). So, by employing a comprehensive and multidisciplinary stepwise experimental design [in vitro assays (several cell lines, high-throughput chemical characterization) + in silico studies (foodinformatics)], this study documents the differential anticancer potential of three PMG fruits of different flesh (aryl) colors [red (cv. Wonderful), pink (cv. Molar de Elche), white (cv. Indian)], although such anticancer potential could be more effective in nutraceutical formulations such as fruit concentrates. The practical application of the evidence reported here may result in the formulation of novel PMG fruit-based OTC nutraceuticals, highlighting their phenogenotype (varietal)-specific phytochemical value as a market differentiation element, increasing the PMG fruit market share currently held by PMG (’Wonderful’) dry concentrates producing reconstituted juices and medicinal syrups.

## Figures and Tables

**Figure 1 biomolecules-12-01649-f001:**
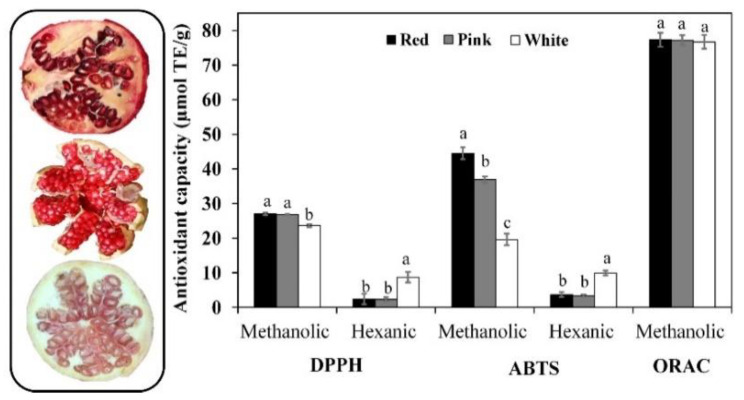
Antioxidant capacity of red (CV. Wonderful), pink (cv. Molar de Elche), and white (cv. Indian) pomegranates fruits. Results are expressed as mean [μmol of Trolox equivalents (TE)/g of dried fruit (methanolic extract) or seed oil (hexane extract)] ± standard deviation (SD). Different letters for the same antioxidant assay signified statistical differences (*p* < 0.05). 2,2′-Azino-bis(3-ethylbenzothiazoline-6-sulfonic acid (ABTS), 2,2-diphenyl-1-picrylhydrazyl (DPPH), Oxygen radical absorbance capacity (ORAC) are displayed.

**Figure 2 biomolecules-12-01649-f002:**
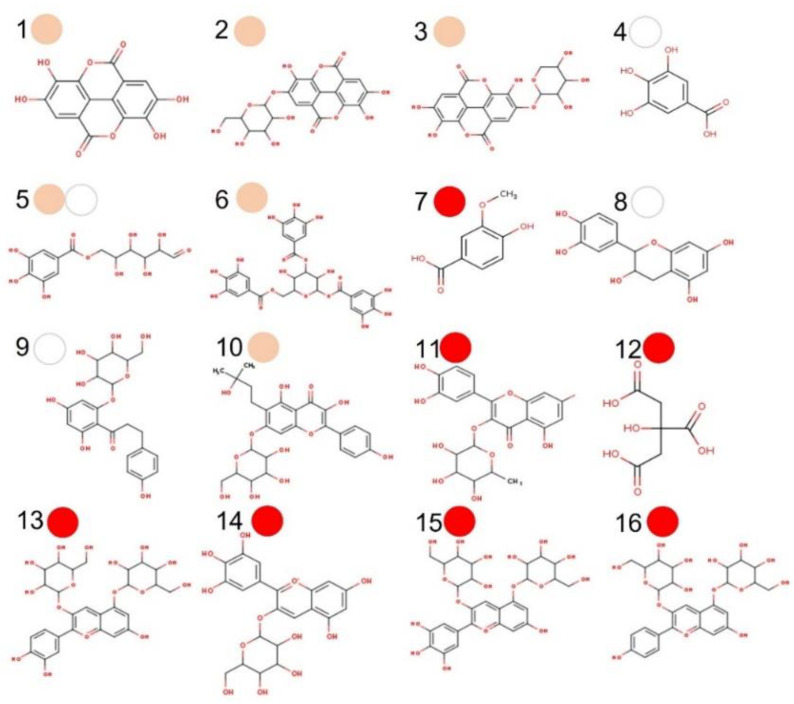
Major phytochemicals in pomegranate (PMG) fruits (aryls) of different color. Ellagic acid (**1**), ellagic acid glucoside (**2**), ellagic acid-4-O-xylopiranoside (**3**), gallic acid (**4**), galloyl-6-O-glucoside (**5**), 1,3,6-tri-O-galloyl-D-glucose (**6**), vanilic acid (**7**), D-(+)-catechin (**8**), phlorizin (**9**), phellatin (**10**), quercitrin (**11**), citric acid (**12**), cyanidin-3,5-O-glucoside (**13**), delphinidin-3-O-glucoside (**14**), delphinidin-3,5-O-diglucoside (**15**), and pelargonidin-3,5-O-diglucoside (**16**). The major PMG source is depicted as a correspondingly colored circle (red, pink, white).

**Figure 3 biomolecules-12-01649-f003:**
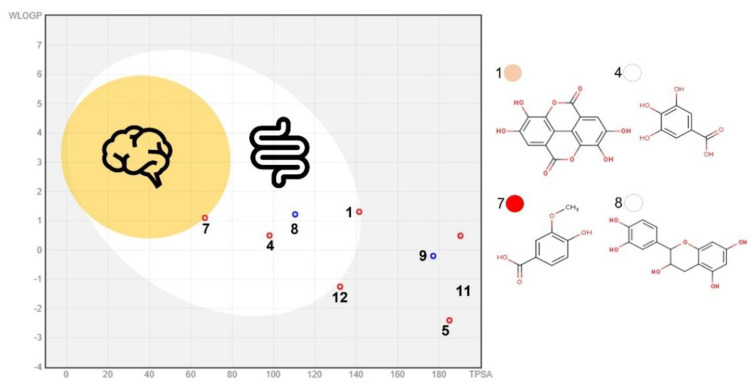
BOILED−Egg model for major pomegranate fruit phytochemicals. The blood−brain barrier (BBB, orange circle) and gastrointestinal (GI, white ellipse) and non-BBB/GI permeants (grey rectangle). See Figure 2 for phytochemical IDs. The model was generated in a SwissADME predictor server (http://www.swissadme.ch/index.php; accessed date: 15 August 2022) [23].

**Figure 4 biomolecules-12-01649-f004:**
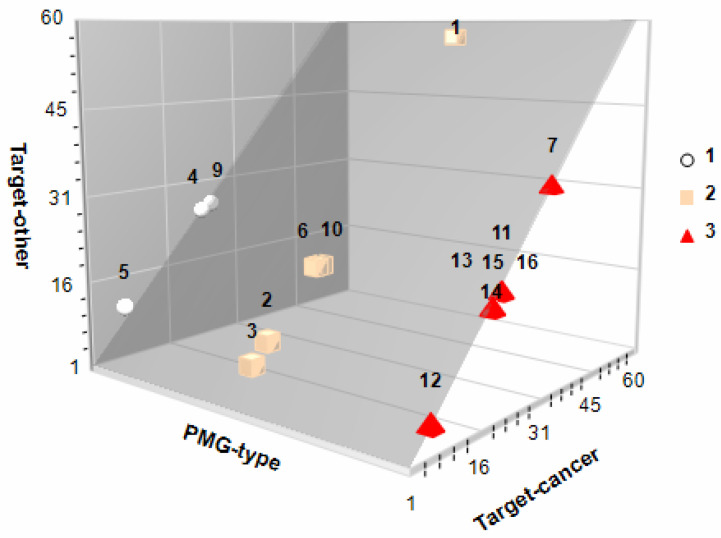
In silico bioactivity potential of pomegranates. 3D scatter plot showing the cancer (*X*-axis) and other (*Y*-axis) protein target prediction for all identified phytochemicals (see Figure 2 for coded phytochemicals) in white (circle), pink (cubes) and red (pyramids) PMGs. Protein targets were identified (SwissTargetPrediction 2019 [59]; http://www.swisstargetprediction.ch/; accessed date: 15 August 2022) and confirmed with UniProt [25] and Pharos [26]).

**Figure 5 biomolecules-12-01649-f005:**
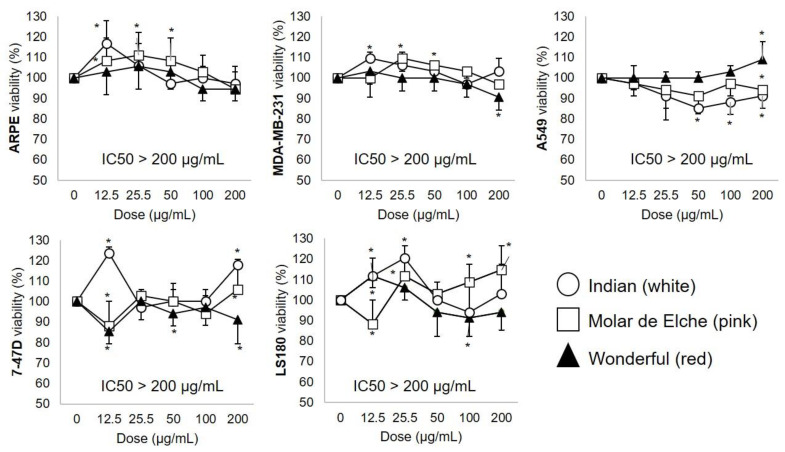
MTT cell viability assay after exposure to graded doses (0–200 μg.mL^−^^1^ in DMSO; 48 h) of PMG fruit extracts (red, pink, white). Data are expressed as mean (triplicate) ± SD. * Statistically different (*p* < 0.05; one-way ANOVA, Tukey’s post hoc test) vs. unstimulated (0 μg.mL^−^^1^) cells; human normal retinal (ARPE-19; CRL-2302TM), breast MDA-MB-231; HTB-26TM), alveolar (A549; CCL-185TM) and colorectal Duke’s type II (LS180; CL-187TM) adenocarcinomas, breast ductal carcinoma (T-47D; HTB-133TM) cell lines (ATCC^®^, Rockville, MD, USA).

**Table 1 biomolecules-12-01649-t001:** Chemical and phytochemical composition of pomegranates of different color. ^1^.

	Red	Pink	White
Moisture (g)	11.8 ± 0.0 ^c^	17.8 ± 0.1 ^a^	14.0 ± 0.7 ^b^
Protein (g)	6.5 ± 0.3 ^b^	7.4 ± 0.4 ^a^	7.3 ± 0.4 ^a^
Fats (g)	0.4 ± 0.0 ^b^	0.9 ± 0.0 ^a^	0.4 ± 0.1 ^b^
Ash (g)	2.4 ± 0.0 ^b^	3.1 ± 0.1 ^a^	2.9 ± 0.0 ^a^
Carbohydrates (g) ^2^	79.0 ± 0.4 ^a^	70.8 ± 0.4 ^c^	75.4 ± 0.1 ^b^
Phenolic compounds (mg GAE)	2450 ± 90 ^a^	2240 ± 30 ^b^	1820 ± 30 ^c^
Flavonoids (mg QE)	320 ± 30 ^a^	250 ± 20 ^b^	170 ± 10 ^c^
Anthocyanins (mg Cy3GE) ^3^	8070 ± 20 ^a^	460 ± 20 ^b^	--
Ascorbic acid (μg)	54.7 ± 12.5 ^b^	18.5 ± 5.4 ^c^	105.2 ± 5.3 ^a^
Total carotenoids (μg)	31.0 ± 1.0 ^a^	32.0 ± 1.2 ^a^	21.0 ± 0.3 ^a^
*Lutein*	31.0 ± 1.0 ^a^	32.0 ± 1.2 ^a^	9.8 ± 0.3 ^b^
*α-Carotene*	--	--	1.5 ± 0.2
*β-Carotene*	--	--	9.7 ± 0.4
Total tocols (μg)	32.1 ± 0.2 ^c^	69.4 ± 1.0 ^b^	110.0 ± 0.2 ^a^
*α-tocopherol*	3.5 ± 0.0 ^a^	3.2 ± 0.0 ^b^	3.4 ± 0.0 ^a^
*γ-tocopherol*	24.7 ± 0.1 ^c^	63.7 ± 2.9 ^b^	103.1 ± 0.1 ^a^
*δ-tocopherol*	3.2 ± 0.2	--	--
*β-tocotrienol*	0.44 ± 0.1 ^c^	2.5 ± 0.1 ^b^	3.1 ± 0.0 ^a^
*γ-tocotrienol*	0.24 ± 0.0 ^a^	--	0.3 ± 0.0 ^a^

Data are expressed as mean (per 100 g DW) ± SD (*n* ≥ 6). ^1^ Different superscript letters (^a^,^b^,^c^) within a line indicate statistical differences (*p* < 0.05); ^2^ by weight difference. Below detection limit/quantification (--); cyanidin-3-glucoside (C3GE), gallic acid (GAE) ^3^ and quercetin (QE) equivalents.

## Data Availability

Not applicable.

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
