# Peer review of "Potential Anticancer Activity of Pomegranate (Punica granatum L.) Fruits of Different Color: In Vitro and In Silico Evidence"

_biomolecules, 2022, doi:10.3390/biom12111649_

Round 1

Reviewer 1 Report

The manuscript of  Cortez-Trejo  et al. describes the anti-cancer effects and phenogenotype of Pomegranate fruits both in vitro and in silico. Furthermore, this study revealed the anticancer potential of the PMG fruit is phenogenotype-specific, although it could be more effective in nutraceutical formulations. The study is well designed and the manuscript is well organized and well written and fairly easy for the reader to follow. Although there are several typos and the narrative is a kind of fragmentary, I just abide by scientific soundness. I would like to offer the following minor points for consideration by the authors towards the improvement of the manuscript:

1- There are a number of grammatical/spelling errors which need correcting. Some examples include:

-P9 Line 385:  Please correct “Bioavailability radars enables a first glance … ”

-P9 Line 387: Please correct “However none of all sixteen molecules…”

-P13 Line 480:  Please correct “In this study in vitro assays indicated...”

2- Please check the correctness of the order of the subtitles in the materials and methods section ( 2.8. and 2.9.)  
3-Finally, some recent and relevant studies could enrich your manuscript.

https://doi.org/10.3892/ijfn.2021.16

https://doi.org/10.1080/10408398.2021.1976721

Author Response

Thank you very much for your observations. The point-by-point response to your requirements is attached as a PDF file

Reviewer 2 Report

Cortez-Trejo et al., reported “Potential Anticancer Activity of Pomegranate (Punica granatum 2 L.) Fruits of Different Color: In vitro and in silico evidence” is complete and well within the scope of this journal. This study describes three phenotypically distinct types of pomegranate fruit, its secondary metabolite profile, and its anticancer potential in a range of cancer cell lines. This study is interesting, well-organized, and effectively written. However, some minor issues must be addressed before to acceptance of this MS.

Comments

1.     In vitro and in vivo are not italicised elsewhere in the manuscript.

2.     The authors should provide pictures of cell cytotoxicity assays.

Author Response

(The authors gave the same response as above.)

Reviewer 3 Report

This study aimed to associate the chemical/phytochemical profile of three Pomegranate fruits of different flesh (aryl) colors [red (cv. Wonderful), pink (cv. Molar de Elche), white (cv. Indian)] with their in vitro/ in silico anti-cancer potential. The comprehensive [in vitro (several cell lines, high-throughput chemical characterization) plus in silico (cheminformatics)] screening for the genotype-specific anticancer potential of PMG fruits. Although this MS has overall interest and visibility, some aspects should still be considered to improve the quality and objectiveness.

1)      Please speculate about the reasons for the obtained results. The discussion needs to improve.

2)      In Conclusion, the authors should add the potential practical application.

3)      The article should be reviewed for English language proficiency and grammar. There are a lot of sentences without sense, misspelled words, and punctuation errors.

Author Response

(The authors gave the same response as above.)
